# CDKL5 Deficiency Disorder: Revealing the Molecular Mechanism of Pathogenic Variants

**DOI:** 10.3390/ijms26178399

**Published:** 2025-08-29

**Authors:** Shamrat Kumar Paul, Shailesh Kumar Panday, Luigi Boccuto, Emil Alexov

**Affiliations:** 1Department of Physics and Astronomy, College of Science, Clemson University, Clemson, SC 29634, USA; shamrap@clemson.edu (S.K.P.); spanday@clemson.edu (S.K.P.); 2Medical Biophysics Graduate Program, College of Science, Clemson University, Clemson, SC 29634, USA; lboccut@clemson.edu; 3Healthcare Genetics and Genomics Interdisciplinary Doctoral Program, School of Nursing, College of Behavioral, Social and Health Sciences, Clemson University, Greenwood, SC 29646, USA

**Keywords:** CDKL5, 3D structure model, CDKL5 deficiency syndrome, in silico modeling, pathogenic variant, missense mutation, folding free energy, binding free energy

## Abstract

Cyclin-dependent kinase-like 5 (CDKL5) deficiency disorder, which is a developmental and epileptic encephalopathy occurring in 1 in every 40,000 to 60,000 live births, was the subject of this computational investigation. This study provided a comprehensive list of missense variants (156) seen in the human population within the CDKL5 protein. Furthermore, the list of CDKL5 binding partners was updated to include four new entries. Computational modeling resulted in 3D structure models of twenty-four CDKL5-target protein complexes. The CDKL5 stability changes upon the above-mentioned missense mutations that were modeled, and it was shown that the corresponding folding free energy changes (ΔΔG_folding_) caused by pathogenic variants are much larger than the ΔΔG_folding_ caused by benign variants. The same observation was made for the binding free energy change (ΔΔG_binding_). This resulted in a protocol that allowed for the reclassification of missense variants with unknown or conflicting significance into pathogenic or benign. It was demonstrated that such reclassification is more reliable than using leading tools for pathogenicity predictions, since the latter failed to correctly predict known pathogenic/benign variants. Furthermore, the study demonstrated that pathogenicity is linked with the disturbance of thermodynamics quantities such as ΔΔG_folding_ and ΔΔG_binding_, paving the way for development of therapeutic solutions.

## 1. Introduction

Cyclin-dependent kinase-like 5 (CDKL5) deficiency disorder (CDD), catalogued in online mendelian inheritance in man (OMIM ID: 300203, 300672) [1], is a severe neurodevelopmental disorder that is also known as early infantile epileptic encephalopathy, which is classified as a developmental and epileptic encephalopathy (DEE) [2,3]. CDD is estimated to affect approximately 1 in every 40,000 to 60,000 live births [4,5,6,7] and arises from pathogenic variants in the *CDKL5* gene, resulting in the production of a nonfunctional protein [8]. This gene, also known as *serine threonine kinase 9* (*STK9*), is located on the X chromosome at position at position Xp22.13 [9].

Although it was originally classified as an early-onset seizure subtype of Rett syndrome, the current understanding recognizes CDD as a separate and distinct neurodevelopmental disorder [10]. Females are affected more frequently than males, with an estimated female-to-male ratio of 4:1 [11,12]. However, the clinical severity of CDD can be comparable between heterozygous females and hemizygous males, and in some cases, males may exhibit more severe symptoms [13,14].

The clinical presentation of CDD encompasses a wide spectrum of severe neurological impairments, with early-onset, drug-resistant epilepsy serving as a defining feature [15,16]. Seizures typically emerge within the first 2–3 months of life and are frequently unresponsive to conventional antiepileptic therapies [17,18]. Features of seizure in CDD commonly include epileptic spasms and tonic seizures [5,15], while less frequent types encompass clonic, atonic, absence, and hypermotor–tonic–spasm sequence episodes [19,20]. Severe global developmental delay and intellectual disability are observed in all individuals with CDD, typically becoming evident within the first months of birth [21,22]. Additional prominent features include motor disturbances such as hypotonia, chorea, dystonia, and stereotyped hand and leg movements, with only a small subset of patients achieving independent ambulation [2,3,5,23,24]. Cortical/cerebral visual impairment (CVI) is commonly observed in individuals with CDD [25,26]. Autonomic dysfunction is also prevalent, including sleep disturbances, breathing irregularities such as apnea and hypoventilation, and gastrointestinal issues that often necessitate gastrostomy tube placement [5,15,27,28]. Musculoskeletal abnormalities, such as scoliosis, have been reported in a subset of patients [29]. Additionally, many individuals exhibit altered pain perception [30]. Although neuroimaging is frequently unremarkable, some cases reveal delayed myelination of mild cerebellar atrophy [31].

The CDKL5 protein belongs to the CMGC kinase group and serves as a key player in cellular signaling pathways, encompassing cell-cycle regulation, proliferation, differentiation, apoptosis, and gene expression regulation [32,33]. Reported *CDKL5* variants include missense variants, nonsense variants, frameshift variants, deletions, truncations, splice variants, and intragenic duplications, with hundreds of known pathogenic variants identified [34]. Most cases of CDD are typically caused by de novo variants, arising either in the germline or post-zygotically after fertilization. A whole-genome sequencing study of 197 patient–parent trios with DEE [35] identified a genetic diagnosis in 63 individuals, 84% of whom carried de novo variants, including variants in *CDKL5*, while only 10% had inherited variants, and the remaining 6% of cases were found to be copy number variants (CNVs). Although the study was not specific to CDD, its findings support the observation that inherited *CDKL5* variants are exceptionally rare [35] and typically arise from a heterozygous or mosaic mother. In such cases, the mother carries the variant on one X chromosome (heterozygous) or in a subset of cells due to the postzygotic mutation (mosaicism); clinical symptoms may be absent or mild due to skewed X-chromosome inactivation (XCI) tending to silence the mutant copy of the *CDKL5* gene, yet transmission of the pathogenic allele to offspring remains possible [3,14,32]. The majority of pathogenic missense variants are clustered within the N-terminal catalytic domain, suggesting that the disruption of CDKL5’s kinase function is a key driver of CDD pathogenesis [32,36]. The large C-terminal region of CDKL5 contributes to the regulation of its enzymatic activity, subcellular localization, and protein stability, indicating its functional importance beyond the catalytic domain [37]; however, the number of missense variants found in this region is much smaller (44 out of 156) than in the catalytic domain (112 out of 156).

The catalytic activity of CDKL5 begins with autophosphorylation and progresses to substrate protein phosphorylation. This activity is structurally supported by domains within its N-terminal kinase domain, including an ATP-binding site (residues 19–43) and an S/T kinase active site (residues 131–143) [32]. CDKL5 auto-phosphorylates on its own Y171 residue of its activation sequence (TEY motif; residues 169–171), which is hypothesized to induce its active kinase configuration [38]. This autophosphorylation event is critical to regulating the catalytic activity of CDKL5, though the exact mechanism of this process is unknown. Once activated, CDKL5 phosphorylates its substrates at a defined consensus motif, RPX[S/T][A/G/P/S], with a given preference for serine (85%) over threonine (15%) as a phosphorylation site [38,39,40]. The CDKL5 phosphorylates multiple neuronal substrates, with several substrate-specific phosphorylation sites identified to date. Loss-of-function variants disrupt these phosphorylation substrates, thus disrupting diverse neuronal processes, and additional novel phosphorylation substrates continue to be identified; these are described later, along with their associated functions. Multiple substrates of CDKL5 have been characterized and validated as phosphorylation substrates of the kinase (reviewed in [32,41]). Beyond its catalytic core, CDKL5 contains a MAPK insert site (residue 297) [41] and multiple nuclear trafficking signals, including two nuclear localization signals (NLS1: 312–315; NLS2: 784–789) and a nuclear export signal (NES: 836–845), which may regulate its subcellular distribution and substrate accessibility [32].

A recent study further expands this repertoire by identifying new phosphorylation substrates for CDKL5 [42]. By using a patient-derived male iPSC cell line carrying the CDKL5 variant c.175C>T (resulting in p.Arg59*, where the asterisk (*) denotes a premature stop codon, according to HGVS nomenclature) [43] alongside CRISPR-Cas9 gene-edited isogenic controls (genetically matched), the authors generated human iPSC-derived cortical cells, which recapitulate features of CDD, such as impaired neurite outgrowth and reduced phosphorylation of EB2, a known direct phosphorylation substrate for CDKL5 [42]. Based on this model, an unbiased phosphoproteomic analysis identified GTF2I, PPP1R35, GATAD2A, and ZNF219 as the novel direct phosphorylation substrates of CDKL5 [42].

The goals of this current computational and literature search study are to provide a comprehensive list of missense variants seen in the human population, to curate new CDKL5 phosphorylation substrates, to generate a 3D structure of the corresponding CDKL5-target complexes, to predict the effect of missense variants on *CDKL5* stability and binding, and to classify missense variants with unknown significance seen in the human population.

## 2. Results

### 2.1. CDKL5 Variants, Structure, and Binding Partners

#### 2.1.1. Curation and Structural Mapping of Missense CDKL5 Variants Associated with CDD

Using ClinVar [44], the 1000 Genomes Project (1KGP) [45], a recent work form the literature [46], and gnomAD [47], we compiled a comprehensive set of *CDKL5* variants (Figure 1A). In ClinVar, a search for the keyword “CDKL5” returned 2193 entries: 1517 single-nucleotide variants (SNVs), with the remaining variant types comprising deletions (269), duplications (139), copy-number losses (107), copy-number gains (78), microsatellite (46), insertions (17), indels (16), complex variants (2), and inversions (2). From these 1517 SNVs, 783 variants were found to be annotated with a “missense_variant” consequence. Of those 783 variants, 485 were “missense variants”, 297 were annotated as “missense_variant|intron_variant”, and 1 as “missense_variant|splice_donor_variant”. Filtering further for the “CDKL5 disorder” condition yielded 139 “molecular consequence” entries, from which only 120 missense variants were associated with CDKL5 disorder. Next, the Genome Reference Consortium Human Build 38: GRCh38-mapped X-chromosome VCF format file from The International Genome Sample Resource (IGSR) of the 1KGP was downloaded and using variant effect predictor (VEP) [48], a total of 4480 variants were extracted within the *CDKL5* gene region (chrX:18,425,583–18,663,629). Among 4480 variants, 4163 were “intron_variant”, and 19 were “missense_variant”. After excluding six missense variants in the neighboring RS1 gene and removing one duplicate, twelve unique “missense_variant” were obtained. Four of these overlapped with ClinVar, leaving eight unique missense variants from 1KGP. Reviewing a recent study [46] added further unique variants to the final collected missense variant dataset. Thus, the initial totals were 120 (ClinVar) + 8 (1KGP) + 30 (recent study: [46]) = 158. Then, each variant’s wild-type amino acid position against the human CDKL5 reference protein sequence (Uniprot ID: O76039), discarding two mismatches, to arrive at a final dataset of 156 missense variants, served as the *CDKL5* missense variant dataset. Based on germline classification, these 156 missense variants across the full-length CDKL5 protein include benign (20), likely benign (10), benign/likely benign (15), likely pathogenic (22), pathogenic/likely pathogenic (24), pathogenic (9), conflicting class of pathogenicity (13), and uncertain significance (43). A total of 112 variants are located within the kinase domain, comprising benign (4), likely benign (1), benign/likely benign (12), likely pathogenic (22), pathogenic/likely pathogenic (24), pathogenic (9), conflicting classifications of pathogenicity (10), and uncertain significance (30). Finally, each variant’s allele frequencies were annotated from gnomAD [47] (Figure 1A).

Later, curated variants were mapped onto the CDKL5 kinase domain (PDB ID: 4BGQ; residue 1–302) (Figure 1B). In this representation, residues are color-coded by clinical classification: blue for benign, red for pathogenic, and magenta for variants of uncertain significance. Additional variant categories such as benign/likely benign, likely benign, likely pathogenic, and pathogenic/likely pathogenic are detailed in Figure 1B but were not included in structural mapping. To map potential interaction partners of CDKL5 residue Y171, we employed the “findclash” tool in UCSF Chimera [49]. This identified van der Waals (VDW) contacts using a 4 Å overlap threshold, excluding hydrogen bond contributions. The analysis revealed several interacting residues, including G22, A23, and Y24. Notably, G22 carries two likely pathogenic variants (G22V and G22E), while Y24 is associated with a “pathogenic/likely pathogenic” variant (Y24C). Moreover, Y171 interacts not only with its adjacent TEY motif residues, T169 and E170, but also with several residues harboring variants of uncertain significance, including D135G, D153V, A173D, and T174N, corresponding to the D135, D153, A173, and T174 positions shown in the inset of Figure 1B.

#### 2.1.2. CDKL5 Partners

To investigate the molecular interactions of CDKL5, we carried out a literature search and extrapolated data from recent review articles [32,41]. The goal was not only to identify such interactions, but also to provide the corresponding experimental evidence and to outline the molecular function associated with the interactions. Below, we briefly outline the additional interactions that were identified (all known partners and their functionalities are provided in Appendix A). The results are summarized in Figure 2 and Table 1.

In a recent TiO_2_-enriched, label-free phosphoproteomics study of CDKL5 P.(Arg59*) iPSC-derived neurons versus isogenic controls, four novel CDKL5 phosphorylation targets matched CDKL5’s consensus RPX[S/T][A/G/P/S] motif [39]. These include PP1 regulatory subunit 35 (PPP1R35), General transcription factor II-I (GTF2I), GATA zinc finger domain containing 2A (GATAD2A), and Zinc finger protein 219 (ZNF219) [42]. PPP1R35 was identified as a CDKL5 target (phosphosite S52), which functions as a regulatory subunit of PP1 at centrioles and primary cilia, where it mediates centriole-to-centrosome conversion [50], supports cell-cycle progression [51], and directs ciliogenesis [51,52], which are essential for neurogenesis and neuronal maturation [53]. In parallel, GTF2I (phosphosite: Ser 674) is a multifunctional transcription factor that assembles at immediate–early promoters and regulates axon guidance, calcium signaling, and neuronal apoptosis [54], cell-cycle genes, and differentiation programs [55,56,57,58]. Additionally, two Nucleosome Remodeling Deacetylase (NuRD) complex subunits, GATAD2A (phosphosite: Ser100) and ZNF219 (phosphosite: Ser114), were found to be key players that may function to regulate chromatin remodeling and activity-dependent gene programs central to neuronal plasticity [42].

**Figure 2 ijms-26-08399-f002:**
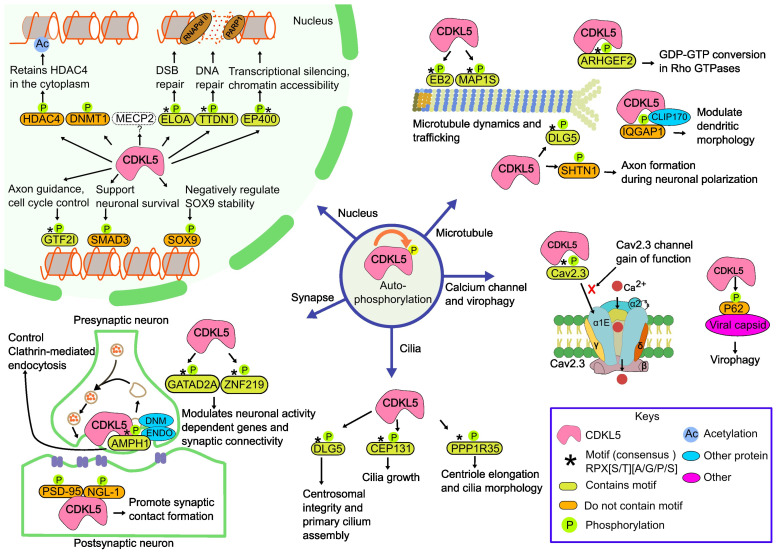
CDKL5-mediated substrate phosphorylation across cellular compartments. CDKL5 orchestrates diverse neuronal processes across distinct subcellular compartments, including cytoplasm, synapse, centrosome, and nucleus. In the cytoplasm, CDKL5 phosphorylates MAP1S, MAPRE2/EB2, ARHGEF2, IQGAP1 (dendritic morphology regulation), and AMPH1 to modulate microtubule dynamics and synaptic vesicle trafficking. Centrosomal/ciliary targets include CEP131, DLG5, and PPP1R35, supporting ciliogenesis and cell-cycle progression. At synapses, CDKL5 binds PSD-95 and NGL-1, influencing dendritic spine formation. In the nucleus, CDKL5 phosphorylates ELOA, EP400, TTDN1, SOX9, GTF2I, GATAD2A, and ZNF219, linking its activity to transcriptional regulation and chromatin remodeling, while the MECP2 phosphorylation mechanism is yet to be explored. Additional targets include Cav2.3 (neuronal excitability), p62 (virophagy), SMAD3, DNMT1, and HDAC4, highlighting CDKL5’s broad role in neuronal homeostasis. The illustration is inspired by the sources ([41,42,59,60]) and other relevant literature cited in the text and sketched using the open-source program Inkscape 1.2.2 [61].

Collectively, these CDKL5 binding partners span distinct cellular compartments and functional categories, suggesting that CDKL5 orchestrates a multifaceted signaling network in neuronal contexts. Disruption of CDKL5 kinase activity is therefore likely to perturb these pathways, contributing to the molecular pathology of CDD.

**Table 1 ijms-26-08399-t001:** Candidate CDKL5 binding partners with consensus motif.

SL	UniProt	Gene	pSite	Consensus Motif (RPX[S/T][A/G/P/S])	Protein	Evidence	Source
1	Q92974	*ARHGEF2*	S122	TIRE**RPSsA**IYPS	Rho guanine nucleotide exchange factor 2	Motif and biochemical	[32]
2	P49418	*AMPH1*	S293	PAPA**RPRsP**SQTR	Amphiphysin1	Motif and biochemical	[32]
3	Q9UPN4	*CEP131*	S35	PVSR**RPGsA**ATTK	Centrosomal protein of 131 kDa	Motif and biochemical	[32]
4	Q8TDM6	*DLG5*	S1115	QKRRR**PKsA**PSFR	Disks large homolog 5	Motif and biochemical	[32]
5	Q14241	*ELOA*	S311	EENR**RPPsG**DNAR	Elongin A	Motif and biochemical	[32]
6	Q96L91	*EP400*	S729	SPVN**RPSsA**TNKA	EE1A-binding protein p400	Motif and biochemical	[32]
7	Q66K74	*MAP1S*	S871, S900	KAPA**RPSsA**SATP, DRAS**RPLsA**RSEP	Microtubule-associated protein 1S	Motif and biochemical	[32]
8	Q15555	*EB2/MAPRE2*	S222	STPS**RPSsA**KRAS	Microtubule-associated protein RP/EB family member 2	Motif and biochemical	[32]
9	Q8TAP9	*TTDN1*	S40	GGGP**RPPsP**RDGY	TTD non-photosensitive 1 protein	Motif and biochemical	[32]
10	P26358	*DNMT1*		N/A	DNA methyltransferase 1	Biochemical	[32]
11	P56524	*HDAC4*	S632	RPLSRAQsSPASAtF	Histone deacetylase 4	Motif and biochemical	[32]
12	Q9HCJ2	*NGL-1/KIAA1580/LRRC4C*	S631	PLLIRMNsKDNVQET	Netrin-G ligand-1	Motif and biochemical	[32]
13	P84022	*SMAD3*	N/A	N/A	Mothers against decapentaplegic homolog 3	Biochemical	[32]
14	P48436	*SOX9*	S199	ATEQTHIsPNAIFKA	Transcription factor SOX-9	Biochemical	[32]
15	P46940	*IQGAP1*	N/A	N/A	IQ Motif Containing GTPase Activating Protein 1	Biochemical	[32]
16	P51608	*MeCP2*	N/A	N/A	Methyl-CpG binding protein 2	Biochemical	[32]
17	P78352	*PSD95/DLG4*	N/A	N/A	Postsynaptic density protein 95	Biochemical	[32]
18	A0MZ66	*SHTN1/SHOT1*	N/A	N/A	Shootin1	Biochemical	[32]
19	P78347	*GTF2I*	S674	QSPK**RPRsP**GSNS	General transcription factor II-I	Motif and biochemical	[42]
20	Q8TAP8	*PPP1R35*	S52	SLSP**RPDsP**QPRH	Protein phosphatase 1 regulatory subunit 35	Motif and biochemical	[42]
21	Q86YP4	*GATAD2A*	S100	KSER**RPPsP**DVIV	GATA zinc finger domain containing 2A	Motif and biochemical	[42]
22	Q9P2Y4	*ZNF219*	S114	HQPE**RPRsP**AARL	Zinc finger protein 219	Motif and biochemical	[42]
23	Q15878	*CACNA1E/Cav2.3*	S14	AVVA**RPGsG**DGD	Voltage-dependent R-type calcium channel subunit alpha-1E (Cav2.3)	Motif and biochemical	[62]
24	Q13501	*SQSTM1/p62*	T269/S272	RSRLTPVsPESS, GGKRSRLtPVSP	Sequestosome-1(p62)	Biochemical	[63]

These data were compiled from the following literature: [32,41,42]. N/A: Not Available.

### 2.2. Homology Modeling of the CDKL5 Kinase Domain and CDKL5-Target Complex Prediction Using ColabFold and CDKL5-Target Protein–Protein Docking Using HADDOCK3

All available CDKL5 structures in the protein data bank, including 4BGQ (resolution: 2.00 Å), 8CIE (resolution: 2.20 Å), and 9EPU (resolution: 2.60 Å), capture the N-terminal kinase domain (residues 1–302). The more recent structures, 8CIE and 9EPU, are co-crystallized with selective small-molecule inhibitors (YL-354 and CAF-382, respectively). We selected 4BGQ for the downstream application because of its higher resolution, 2.00 Å, [64] and the absence of large conformational shifts that would hinder protein–protein interaction modeling. While 4BGQ includes two engineered phosphomimetic variants, T169D and Y171E, which were introduced to mimic phosphorylation and promote an active kinase conformation [49], these positions were reverted to the wild-type residues, Thr169 and Y171, during the homology modeling using Modeller 10.4 [65]. The resulting model was superimposed on the original 4BGQ structure to validate the structural fidelity using UCSF Chimera [49]. The root mean square deviation (RMSD) between 276 pruned atom pairs was 0.268 Å, indicating strong preservation of the native fold [65] (Figure 3A).

We applied ColabFold 1.5.5, a high-throughput adaptation of AlphaFold-Multimer [66], to predict the complex of CDKL5 with its target proteins (Table 1) and systematically evaluated the spatial relationship between CDKL5 residue Y171 and each substrate’s known phosphorylation site (Figure 3B(1–18)). Y171 lies within the conserved TEY activation motif of CDKL5 and is critical for catalytic activity, so close proximity to a substrate’s phosphorylation site could be a strong indicator of a viable phosphorylation event. Therefore, all ColabFold-generated complexes were visualized in UCSF Chimera [49], and minimum distances between the Y171 (hydroxyl oxygen) and the known phosphosite, Serine (hydroxyl group), of each substrate were measured and visually inspected. Across the full set of substrates (Figure 3B(1–18) and Table 1), three complexes exhibited particularly short Y171-phosphoserine distances, suggesting potential for direct phosphorylation: CDKL5-AMPH1, CDKL5-SOX9, and CDKL5-GATAD2A (Figure 3B(2,11,17)). Upon visual inspection of each substrate’s phosphorylation loop, most lacked a well-defined fold, except SOX9 and ZNF219, both of which possessed an alpha-helix adjacent to the phosphosite (Figure 3C(III,IV)). Therefore, based on both close proximity and a properly folded phosphoserine region, we selected CDKL5-AMPH1, CDKL5-SOX9, CDKL5-GATAD2A, and, additionally, CDKL5-ZNF219 for our downstream studies, as shown in Figure 3C(I–IV).

Upon CDKL5-target protein complex modeling using ColabFold, the HADDOCK3 docking was performed for CDKL5 in complex with each selected substrate, using phosphosite-centered ambiguous interaction restraints (AIRs). In all models, the phosphoserine residue was consistently positioned within hydrogen-bonding distance (2.7–3.3 Å) of CDKL5’s catalytic Y171, confirming a well-posed docking into the phosphosite (Figure 4A–D).

To compute binding energetics, we extracted per-model energy terms from HADDOCK3’s CAPRI output and computed the mean ± SD for six metrics: vdW, Elec, Desolv, AIR, total energy, and HADDOCK score (Appendix A). Among the generated complexes, CDKL5-SOX9 and CDKL5-ZNF219 demonstrated the most favorable non-bonded interactions and, at the same time, interacted with CDKL5 Y171 residue; thus, they were appended to the structural models predicted with ColabFold [66].

### 2.3. Folding, Docking ΔΔG_folding_ and ΔΔG_binding_ Analysis, and Variant Reclassification

#### 2.3.1. Folding Free Energy Change (ΔΔG_folding_)

To assess the impact of missense variants on CDKL5 protein stability, we computed the change in the folding free energy (ΔΔG_folding_) using ten prediction tools (five for sequence-based, and five for structure-based; see Section 4.3) and compared their distributions across benign and pathogenic *CDKL5* missense variants. Analyses were performed independently for the full-length protein for the sequence-based methods (residue 1–960) and the kinase domain for structure-based methods (residue 1 to 302; Figure 5 and Appendix A).

Across the full-length protein (residues 1–960; Figure 5A), pathogenic variants consistently exhibited more negative ΔΔG_folding_ values, indicating greater destabilization as compared to the benign variants. This trend was observed across all sequence-based methods. For example, I-Mutant2.0 predicted a mean of −1.332 kcal/mol for pathogenic variants compared to −0.800 kcal/mol for benign ones. DDGemb showed the strongest separation with means of −1.364 and −0.114 kcal/mol for pathogenic and benign variants, respectively. SAAFEC-SEQ followed a similar pattern, with pathogenic variants averaging −1.341 kcal/mol and benign ones averaging −0.948 kcal/mol. INPS showed a moderate shift between benign and pathogenic classes, while DDGun shows minimal separation, with benign variants centered near zero and pathogenic variants exhibiting high variance. Focusing on the kinase domain (residues 1 to 302), which is functionally critical, the separation between variant classes became more pronounced. Among sequence-based tools (Figure 5B), I-Mutant2.0 predicted a mean ΔΔG_folding_ of −1.332 kcal/mol for pathogenic variants, and −0.183 kcal/mol for benign ones. DDGemb showed a similar distinction, with means of −1.364 and −0.030 kcal/mol, respectively. SAAFEC-SEQ predicted −1.341 kcal/mol for pathogenic variants and −0.870 kcal/mol for benign ones. INPS showed a moderate difference, while DDGun failed to distinguish between classes.

Structure-based tools provided even clearer class distinctions. In the full-length context (Figure 5C), I-Mutant2.0 predicted a mean ΔΔG_folding_ of −1.330 kcal/mol for pathogenic variants and +0.002 kcal/mol for benign ones. mCSM showed a similar trend, with means of −1.025 and −0.258 kcal/mol for pathogenic and benign variants, respectively. INPS predicted means of −0.928 kcal/mol for pathogenic variants and +0.100 kcal/mol for benign ones. DDMut also separated the classes effectively, while DDGun remained neutral for benign variants and destabilizing for pathogenic ones. Structure-based methods delivered the strongest separation in the kinase domain (Figure 5D). mCSM predicted a mean ΔΔG_folding_ of −1.025 kcal/mol for pathogenic variants and −0.258 kcal/mol for benign ones. I-Mutant2.0 (structure) maintained its strong performance, with predictions of −1.330 and +0.002 kcal/mol for pathogenic and benign variants, respectively. INPS predicted −0.928 kcal/mol for pathogenic variants and +0.100 kcal/mol for benign ones. DDMut showed a similar pattern, while DDGun remained neutral for benign variants and destabilizing for pathogenic ones.

These results demonstrate that structure-based ΔΔG_folding_ predictors offer the most reliable separation between pathogenic and benign *CDKL5* variants (Figure 5D) compared to the sequence-based methods. I-Mutant2.0 (structure) and mCSM emerged as the most discriminative tools between benign and pathogenic variants. In cases where structural models are unavailable, I-Mutant2.0 (sequence-based), DDGemb, and SAAFEC-SEQ provide suitable alternatives for predicting variant pathogenicity.

#### 2.3.2. Binding Free Energy

We used the structural models of CDKL5 predicted by ColabFold [66] bound to the corresponding target (SOX9 (197–202), AMPH1 (290–294), GATAD2A (97–101), and ZNF219 (111–115), and carrying benign and pathogenic *CDKL5* variants on the CDKL5 kinase domain. We used four structure-based ΔΔG_binding_ predictors (DDMutPPi, iSee, mCSM-PPI, and SAAMBE-3D) to compute the impact of *CDKL5* variants on ΔΔG_binding_ of selected CDKL5-target protein complexes. Note that not all predictors follow the same ΔΔG_binding_ conventions. iSEE and SAAMBE-3D report values as ΔΔG_binding_ = ΔG_mutant_ − ΔG_wildtype_, while mCSM-PPI and DDMutPPI use the opposite definition (ΔΔG_binding_ = ΔG_wildtype_ − ΔG_mutant_); trends are therefore interpreted consistently as pathogenic variants showing greater destabilization than benign, independent of sign.

For CDKL5-SOX9 (phosphomotif: 197–202) (Figure 6, Column 1 from left), iSEE predicted a mean ΔΔG_binding_ of 1.86 kcal/mol versus 1.25 kcal/mol for benign, indicating reduced binding affinity for the pathogenic variants. mCSM-PPI showed a similar trend, with pathogenic variants averaging −0.91 kcal/mol and benign variants −0.53 kcal/mol. DDMutPPI predicted more negative ΔΔG_binding_ for pathogenic variants (−0.53 kcal/mol) compared to benign (−0.13 kcal/mol); similarly, SAAMBE-3D showed minimal separation (0.44 vs. 0.22 kcal/mol). For CDKL5-AMPH1 (phosphomotif residues: 290–294), pathogenic variants showed a substantial increase in ΔΔG_binding_ under iSEE (2.38 kcal/mol) compared to benign (1.35 kcal/mol), suggesting strong binding disruption. mCSM-PPI predicted more negative ΔΔG_binding_ for pathogenic variants (−0.90 kcal/mol) versus benign (−0.55 kcal/mol). DDMutPPI and SAAMBE3D showed smaller shifts (−0.30 vs. −0.07 kcal/mol and 0.29 vs. 0.15 kcal/mol, respectively). For CDKL5-GATAD2A (phosphomotif residues: 97–101), iSEE again showed strong class separation, with pathogenic variants averaging 2.56 kcal/mol and benign variants 1.26 kcal/mol. mCSM-PPI predicted −0.73 kcal/mol for pathogenic and −0.50 kcal/mol for benign variants. DDMutPPI showed a modest shift (−0.42 vs. −0.13 kcal/mol), while SAAMBE-3D yielded minimal separation (0.29 vs. 0.13 kcal/mol). For CDKL5-ZNF219 (phosphomotif residue: 111–115), iSEE predicted a ΔΔG_binding_ of 2.20 kcal/mol for pathogenic variants versus 1.02 kcal/mol for benign. mCSM-PPI showed a similar pattern (−0.74 vs. −0.52 kcal/mol), and DDMutPPI predicted −0.48 kcal/mol for pathogenic versus −0.11 kcal/mol for benign. SAAMBE-3D again showed limited separation (0.35 vs. 0.17 kcal/mol). Overall, iSEE consistently produced the largest ΔΔG_binding_ shifts between benign and pathogenic variants across all four complexes, supporting its usefulness in identifying binding-disruptive variants. mCSM-PPI also demonstrated reliable separation, particularly in detecting destabilizing effects of pathogenic variants. DDMutPPI offered moderate sensitivity, while SAAMBE-3D showed minimal discriminative power.

#### 2.3.3. Folding Threshold

To determine a reliable stability-based criterion for classifying *CDKL5* missense variants, we analyzed structure-based ΔΔG_folding_ predictions across five computational methods. For each variant, we computed folding free energy changes (ΔΔG_folding_) from five different methods and picked the maximum value across methods, and thus we find ΔΔG_Fmax_ for each variant. The four benign variants consistently showed low maximum absolute ΔΔG_Fmax_ values, ranging from 0.09 to 0.68 kcal/mol (Figure 7(left)). In contrast, the nine pathogenic variants exhibited higher ΔΔG_Fmax_ values, spanning from 0.86 to 3.42 kcal/mol (Figure 7(middle)). By selecting the midpoint between the largest benign and smallest pathogenic ΔΔG_Fmax_ values, we defined a threshold of 0.77 kcal/mol (Figure 7 (right)). This cutoff separated the two classes, with all variants correctly classified according to their clinical annotation.

#### 2.3.4. Binding Threshold

Firstly, for each CDKL5-target protein complex and each variant, we calculated the mean absolute ΔΔG_binding_ across four prediction methods (DDMutPPI, iSEE, mCSM-PPI, and SAAMBE-3D). Benign *CDKL5* variants induced only modest destabilization of partner binding interfaces, with mean absolute ΔΔG_binding_ values across the four methods of 0.42–0.82 kcal/mol for CDKL5-AMPH1 (motif 290–294; Figure 8(A1)), 0.45–0.58 kcal/mol for CDKL5-GATAD2A (motif 97–101; Figure 8(A2)), 0.40–0.58 kcal/mol for CDKL5-SOX9 (motif 197–202; Figure 8(A3)), and 0.29–0.77 kcal/mol for CDKL5-ZNF219 (motif 111–115; Figure 8(A4)); in each case, the iSEE predictor reported the highest shifts (up to ~1.25 kcal/mol), while the other methods remained below ~0.70 kcal/mol. In contrast, pathogenic variants caused substantially larger perturbations, with complex-average |ΔΔG_binding_| values of 0.91–1.55 kcal/mol for CDKL5-AMPH1 (motif 290–294; Figure 8(A5)), 0.93–1.47 kcal/mol for CDKL5-GATAD2A (motif 97–101; Figure 8(A6)), 0.95–1.38 kcal/mol for CDKL5-SOX9 (motif 197–202; Figure 8(A7)), and 0.89–1.32 kcal/mol for CDKL5-ZNF219 (motif 111–115; Figure 8(A8)), again driven primarily by elevated iSEE predictions. To distill these results into a single metric, for each variant, we defined ΔΔG_Bmax_ as the maximum complex-average |ΔΔG_binding_| across the four CDKL5-partner interactions and plotted its value (Figure 8B). Benign ΔΔG_Bmax_ ranged from 0.45 to 0.82 kcal/mol, whereas pathogenic ΔΔG_Bmax_ spanned from 0.95 to 1.55 kcal/mol; by placing a cutoff at the midpoint (0.88 kcal/mol) between the highest benign and lowest pathogenic value, we achieved discrimination of clinical impact (benign/pathogenic), demonstrating that ΔΔG_Bmax_ can be used as classifier of *CDKL5* variant pathogenicity.

Additionally, to gain better insight into the classification of pathogenicity based upon the above-discussed methodology, we also used folding (ΔΔG_Fmax_) and binding (ΔΔG_Bmax_) free-energy changes to separate benign from pathogenic CDKL5 variants, with label-flip-invariant metrics (AUROC_sym, balanced accuracy, and MCC_sym) indicating predictive trends rather than precise performance due to the small dataset (13 variants: 4 benign and 9 pathogenic; see Appendix A and related description).

#### 2.3.5. Variants Reclassification Based on ΔΔG_folding_ and ΔΔG_binding_ Thresholds

Before reclassification, folding ΔΔG_Fmax_ for the four benign variants ranged from 0.09 to 0.68 kcal/mol, while for nine pathogenic variants, it ranged from 0.86 to 3.42 kcal/mol; a midpoint threshold of 0.77 kcal/mol cleanly separates the two classes (benign/pathogenic) (Figure 9A). After applying this cutoff to total missense variants (112), the benign group grew to 14 (ΔΔG_folding_ range 0.09–0.72), and the pathogenic group to 98 (ΔΔG_folding_ range 0.79–4.2 kcal/mol) (Figure 9B). Figure 9C shows how each original germline category (e.g., “benign/likely benign” and “uncertain significance”) redistributed. For example, 29 “uncertain significance” variants moved to pathogenic and 1 to benign.

Similarly, binding ΔΔG_Bmax_ for benign variants originally spanned 0.45–0.82 kcal/mol, and for pathogenic variants, 0.95–1.55 kcal/mol, with a threshold of 0.88 kcal/mol (Figure 9D). Reclassification of 112 total variants yielded 32 benign (ΔΔG_binding_ range: 0.41–0.88 kcal/mol) and 80 pathogenic (ΔΔG_binding_ range: 0.90–2.14 kcal/mol; Figure 9E). Figure 9F shows that among variants of “uncertain significance”, 18 were reclassified as pathogenic and 12 as benign. Overall, these distributions and reclassification counts demonstrate that both ΔΔG_Fmax_ and ΔΔG_Bmax_ thresholds logically classify *CDKL5* variants by their clinical impact.

#### 2.3.6. Variants Reclassification Based on Pathogenicity Score

In Figure 10, the top-left panel (PolyPhen-2), every one of the 112 recalled variants was called pathogenic; PolyPhen-2 reclassified all 12 “benign/likely benign” and all 30 “uncertain significance” variants as pathogenic, resulting in 100% sensitivity but no specificity. In Figure 10, the top-right panel (MutPred2) shows 29 benign recalls versus 83 pathogenic; it retained the four original benign variants, reclassified 10 of 12 “benign/likely benign” as benign, and split “uncertain significance” into 8 benign and 22 pathogenic.

In Figure 10, the bottom-left panel (ESM-1v), 20 variants were re-established as benign and 92 pathogenic; ESM-1v also preserved all four true benign variants, re-established 8/12 “benign/likely benign” correctly, and categorized 4 of 30 “uncertain significance” as benign. Finally, in Figure 10, the bottom-right panel, AlphaMissense called 14 benign and 98 pathogenic, capturing 3 of 4 original benign variants and 7 out of 12 “benign/likely benign”, but only 3 of 30 “uncertain significance” as benign. Overall, MutPred2 and ESM-1v (top-right and bottom-left in Figure 10) showed the best balance between detecting true positives and avoiding false ones. In contrast, PolyPhen-2 and AlphaMissense (Figure 10: top-left and bottom-right) tended to overpredict pathogenic variants.

Upon application of evolution-based pathogenicity predictors (Polyphen-2, MutPred2, ESM-1v, and AlphMissense) to predict the pathogenicity of curated CDKL5 variants on its kinase domain (n = 112 variants), we found that most of these approaches failed to cleanly separate benign from pathogenic variants. For this, “accuracy” was defined as the proportion of variants correctly classified as benign (“benign”, “benign/likely benign”, and “likely benign”) versus non-benign (“likely pathogenic”, “pathogenic”, “pathogenic/likely pathogenic”, “uncertain significance”, and “conflicting classifications of pathogenicity”). Predictions of “benign” were considered correct only for variants within the benign group; otherwise, predictions of “pathogenic” were considered correct. This binary grouping was chosen to allow for direct comparison across pathogenicity methods, though it systematically favors predictors that force ambiguous categories (“uncertain significance”/“conflicting classification of pathogenicity”) into pathogenic (Appendix A).

Polyphen-2 achieved the highest apparent accuracy (93.8%), but it misclassified 2 known “benign” variants as “pathogenic” and reassigned 29 “uncertain significance” and 10 “conflicting classifications of pathogenicity” variants to the “pathogenic” category. MutPred2 showed 77.7% accuracy, with 1 known pathogenic variant mislabeled as “benign”, and shifted 22 “uncertain significance” variants to “pathogenic”. Notably, ESM-1v correctly classified all strictly annotated variants (benign 4/4; pathogenic 9/9), demonstrating perfect recall on curated “benign” and “pathogenic” cases. However, its overall accuracy was 85.7%, reduced by overcalling among ambiguous categories (26 “uncertain significance” and 9 “conflicting classifications of pathogenicity” reassigned as “pathogenic”). AlphaMissense reached 89.3% accuracy, misclassifying 1 benign variant as pathogenic and forcing 27 “uncertain significance” and 10 “conflicting classification of pathogenicity” cases into “pathogenic”.

Overall, while around 25% (ESM-1v; one out of four methods) predictors perform relatively well on strictly benign/pathogenic variants, they consistently over-predict pathogenicity in ambiguous categories: “uncertain significance” or “conflicting classification of pathogenicity” (Appendix A). This inconsistency motivates the CDKL5 variant reclassification strategy grounded in protein thermodynamics, whereby thresholds of ΔΔG_folding_ and ΔΔG_binding_ are applied to capture variant-induced destabilization of structure and interaction energetics, thereby resolving ambiguous cases on the basis of underlying biophysical principles.

Therefore, pathogenicity predictors sometimes fail to distinguish between variants already annotated as benign or pathogenic in curated databases. Meanwhile, the thermodynamic approach, using folding free energy change ΔΔG_Fmax_ and binding free energy change ΔΔG_Bmax_, was able to clearly separate these known pathogenic and benign variants. Based on this thermodynamic reclassification, on the kinase domain of CDKL5, out of 112 variants, 86 pathogenic variants showed greater folding destabilization (ΔΔG_Fmax_ > ΔΔG_Bmax_). This variant reclassification according to the American College of Medical Genetics and Genomics (ACMG) and the Association for Molecular Pathology (AMP) guidelines is provided in Appendix A. In the CDKL5 kinase domain (Figure 11), these residues are mapped in red spheres and exhibit potential sites for the development of drugs aimed at enhancing protein stability. In contrast, seventeen pathogenic variants show higher or equal binding destabilization (ΔΔG_Fmax_ ≤ ΔΔG_Bmax_). These residues, shown in blue in Figure 11, are the potential targets for developing therapeutic interventions aimed at restoring binding between mutant CDKL5 and its binding partners.

## 3. Discussion

Understanding the molecular mechanism of a disease is crucial for the development of treatment. In the case of CDKL5 deficiency, there are many pathogenic variants and many genotypes that result in the disease. Our study extended the list of known missense variants in CDKL5 and further enriched the list of genotypes, resulting in 156 missense variants in the full-length CDKL5 protein, while 112 missense variants fall within the kinase domain, and these 112 missense variants in the kinase domain are our focus of variant reclassification. Among them, 88.4% (99 out of 112) do not have strict classification as either pathogenic or benign; these variants were re-classified using the methodology described in the manuscript, resulting in 98 pathogenic and 14 benign variants based on ΔΔG_folding_. In parallel, 80 pathogenic and 32 benign variants were reclassified based on ΔΔG_binding_.

A crucial component for any drug discovery is the knowledge of the function and details of the function of the corresponding protein target. To facilitate this, we carried out a literature search and identified four additional partners, which combined with the original review articles [32], resulted in twenty-four interacting partners. Furthermore, structural modeling was carried out to predict 3D structures of the corresponding CDKL5-partner complexes, and four acceptable models were delivered.

While knowledge of the variants in *CDKL5*, both pathogenic and benign, and the knowledge about CDKL5’s function and interacting partners are important for drug development, still one needs to find out what the phenotype is that is caused by the genotypes. Recent works [67,68,69] demonstrated that there is a strong linkage between pathogenicity and thermodynamical properties as folding and binding free energies. Building on these observations, we predicted the folding free energy changes caused by the above-mentioned variants and showed that indeed the pathogenic variants destabilize the CDKL5 protein much more than benign variants (Appendix A). The same was demonstrated for the binding free energy changes caused by the variants. Thus, the study collapsed the genotypes into two phenotypes: changes in folding and binding free energy. This was used to reclassify variants with uncertain significance.

Combining all together, the study suggests that therapeutic solutions for variants (F13S, G20D, G20R, E21G, G22E, G22V, Y24C, V27A, C30Y, R31G, T35I, I41F, K42R, L64P, L67F, L67P, N71D, N71S, I72N, I72T, K76E, R80H, G83V, L97P, V107D, Y117C, L119R, A122T, W125C, C126Y, H127R, V132G, D135G, P138L, L141F, I143N, I143V, H145Y, N146S, K150R, C152F, C152R, D153G, G155D, A157P, A157V, R158H, R158P, R175S, W176C, W176G, W176R, Y177C, Y177S, R178Q, S179F, E181A, L182P, L184H, A186T, D193G, D193H, D193N, G198D, G198R, C199R, L201P, G202E, E203D, E203K, G207E, P209R, G213E, Q219K, Q219P, L220P, K225R, L227R, Y262H, L271P, R285S, T288I, C291R, C291Y, T296A, and L302F) should be sought in developing drug(s) that can enhance mutant CDKL5 stability. For the variants (G20V, G25R, A40V, R59P, R65Q, R80L, H127Y, D153V, V172I, A173D, T174N, R178P, R178W, P180L, D193V, S196L, and G213R), the efforts should be to develop drugs capable of enhancing binding affinity of the mutant CDKL5 protein to the corresponding partner. Such a drug development was demonstrated to be quite successful [70,71], and strategies for carrying out such development are outlined in a recent review [72]. Figure 11 shows the CDKL5 catalytic domain with all pathogenic mutations mapped onto the 3D structure. One can appreciate that mutations that are predicted to affect mostly stability are grouped within several structural regions and can be targeted by the same drug, while mutations affecting mostly the binding affinity are grouped in different parts of the CDKL5 structure and should be targeted with different small molecules, potential drugs. This demonstrated that while the pathogenic mutations are many, their effect can be mitigated with several drugs only.

## 4. Materials and Methods

### 4.1. Data Collection

To compile *CDKL5* missense variants, we first queried the ClinVar [44] database using the keyword “CDKL5”. Next, we obtained the GRCh38-aligned X-chromosome VCF from the 1000 Genome Project (1KGP) from IGSR [73] and used Ensembl’s variant effect predictor v113.0 (VEP) [48] tool to extract *CDKL5*-gene region missense variants, and discarded non-*CDKL5* and duplicate calls. In a recent study [46], a set of 76 *CDKL5* variants was curated for additional unique missense entries. After merging these three resources (ClinVar, 1KGP, literature), we matched each variant’s wild-type amino acid sequence to the CDKL5 reference sequence, discarded mismatches and eliminated overlaps to yield a final, non-redundant set of 156 missense variants. Among these 156 missense variants, 112 are within the kinase domain. Each variant’s corresponding allele frequency was obtained from gnomAD.

### 4.2. CDKL5 Structure Preparation and Prediction of Complex with Its Binding Partners

The 4BGQ structure was processed using Biopython [74] to eliminate non-standard residues while retaining essential crystallographic metadata, including the “CRYST1” record. The complete CDKL5 amino acid sequence from Uniprot was aligned with the SEQRES-based sequence extracted from 4BGQ, and any inconsistencies, including engineered mutations, were resolved. We applied Modeller 10.4 [65], and we reconstructed any missing or altered residues, considering the cleaned 4BGQ [64] as the template structure and the UniProt sequence as the modeling target. This pipeline confirmed a native-like conformation with uninterrupted backbone continuity from residue 1 to 302 (kinase domain), yielding a structure suitable for the downstream analysis. Additionally, we employed ColabFold 1.5.5 [66], a high-throughput adaptation of AlphaFold2, to model the structures of CDKL5 and its binding partners complex. Protein sequences were curated and formatted in FASTA for batch-mode processing.

### 4.3. Folding Free Energy Calculations

To compute the impact of variants (point mutations) in folding free energy on the human CDKL5 protein, we employed several State-of-the-Art computational methods that utilize both sequence and structure information of the protein. Using the UniProt ID O76039, the amino acid sequence of human CDKL5 was obtained, while the kinase domain of the CDKL5 X-ray crystal structure was collected from the Protein Data Bank using the PDB ID 4BGQ. Subsequently, the missing and mutated residues in the X-ray structure were reverted to wild-type using the Modeller 10.4 program [65].

DDGemb [75] is a deep learning-based approach for predicting changes in ΔΔG_folding_ values upon single and multi-point variants using only protein sequence data. It utilizes embeddings generated from the ESM2 [76] protein language model (pLM) and processes them using a Transformer-based neural network. Once it encodes the wild-type and mutant sequences, their residue-level embeddings are then differentiated and applied to predict the change in stability. The model was trained on the S2450 dataset (derived from S2648, derived from ProTherm and FireProtDB [77,78,79]), while on their independent S669 benchmark dataset, it achieved a PCC of 0.68, which outperforms many established methods [75].

DDMut [80] is a structure-based deep-learning framework that predicts folding free energy changes (ΔΔG_folding_) upon single and multiple point variants. It employs a Siamese neural network architecture that utilizes both forward and reverse mutations [80]. It integrates graph-based representations of the local three-dimensional surroundings of the mutated residue with structural and biochemical attributes, including solvent accessibility, residue depth, and atomic interactions [80]. The model processes these features through convolutional and Transformer layers, enabling it to learn both localized and broad mutation impacts [80]. For single-point mutations, they curated their training dataset from S2648, which is originally derived from ProTherm and FireProtDB [77,78,79]. For their multiple point mutations, they prepared their training dataset from the DynaMut2 [81] training set, termed SM1242. They also expanded their dataset by reversing each mutation. DDMut achieved a PCC up to 0.70 on multiple independent blind test sets [80].

Single-amino-acid folding free energy changes SEQ (SAAFEC-SEQ) [82] is a machine-learning method that utilizes sequence-based information to predict folding free energy changes (ΔΔG_folding_) upon single-point mutations [82]. It employs knowledge-based terms and evolutionary information and does not require a 3D structure of the protein [82]. This method uses the gradient-boosting decision tree algorithm. Its features include sequence features (neighbors); physicochemical properties of mutation sites; and evolutionary information, such as Pseudo Position Specific Scoring Matrix (PsePSSM) and neighbor mutation conservation scores [82].

DDGun and DDGun3D [83] are untrained predictors for sequence- and structure-based methods that predict changes in protein folding stability (ΔΔG_folding_) due to single and multiple point mutations. DDGun depends on a linear combination of statistical scores such as BLOSUM62 similarity [84], Skolnick potential, and hydrophobicity differences [83]. It also integrates structural features such as solvent accessibility and the Bastolla-Vendruscolo potential [83]. These weights are then optimized using widely used training datasets S2648 (derived from ProTherm and FireProtDB) [77,78,79], VariBench [85], as well as manually curated datasets. Performance assessments indicate achievement of Pearson correlation coefficients of approximately 0.5 for single-site variants and around 0.5 for the multiple-site variants [83].

Impact of Non-synonymous variations on Protein Stability-Multi-Dimension (INPS-MD) [86] is a method for the prediction of protein stability changes upon single point variation from protein sequence (INPS) and structure (INPS-3D). INPS employed support vector regression (SVR) with radial basis function (RBF) kernels to analyze features including substitution matrices, hydrophobicity indices, and evolutionary conservation data [86]. The structure-based version, INPS-3D, integrates structural descriptors such as relative solvent accessibility and energy difference scores derived from contact potential calculations [86]. Both sequence and structure-based models were trained on the widely used S2648 [77] dataset and benchmarked using blind test sets, which are a subset of the S2648 and a curated P53 mutation dataset [86]. The INPS-3D achieves a Pearson’s correlation of 0.58 in cross-validation, while for the blind tests it scores 0.72. test set, while the sequence-based method INPS performs slightly lower.

mCSM [87] is a structure-based machine learning approach that utilizes a graph-based signature to grasp the geometric and chemical environment of introduced mutations and is capable of predicting the protein folding stability and protein–protein or protein–DNA binding affinity changes. It employs graph-based structural signatures that encode the 3D environment of the mutated residue by measuring the distance patterns between atoms, grouped by pharmacophoric properties (hydrophobicity, charge, hydrogen bonding potential) [87]. These feature representations are used as input to a Gaussian Process regression model trained on different datasets. Using ProTherm-derived S2648, S1925 and S350 datasets [77] the protein folding stability model was trained and achieved a PCC of 0.824 on S1925 and 0.69 on S2648. On the other hand, using SKEMPI and ProNIT datasets, mCSM achieved PCCs of 0.80 for protein–protein affinity changes and 0.67 for protein–DNA affinity changes [87].

Using a Support Vector Machine (SVM)-based method, I-Mutant2.0 [88] was deployed to predict the stability changes due to single amino acid alterations, both at the sequence and structure level. It uses a neural network system to predict the direction in which a mutation affects protein stability, rather than providing a direct ΔΔG_folding_ value [88].

### 4.4. HADDOCK3 Protein–Protein Docking

To investigate the molecular interaction between CDKL5 and its binding partners (e.g., SOX9 and AMPH1), we carried out a systematic protein–protein docking using HADDOCK3 [89], which stands for high ambiguity-driven docking, a versatile and modular platform for integrative structural modeling of bimolecular complexes. HADDOCK3 is fundamentally a data-driven docking platform that integrates experimental, predicted, or inferred interaction information to guide biomolecular complex formation [89]. Its standard workflow comprises four main modules: “topoaa” for generating topology files, “rigidbody” for initial rigid-body docking and sampling, “flexref” for refining the interface with limited flexibility, and “emref” for performing energy minimization in solvent [89]. These modules are customizable, as they can be reordered or skipped depending on the modeling objective [89]. To conduct docking towards biologically relevant interfaces, HADDOCK3 allows the application of spatial restraints, including Ambiguous Interaction Restraints (AIRs), which permit flexible pairing among sets of potential interface residues, and unambiguous restraints, which enforce specific residue–residue contacts, ensuring guided sampling around plausible interface regions [89].

In this study, we prepared the CDKL5 structure (PDB ID: 4BGQ) by rebuilding missing residues and reverting mutated residues (described in the CDKL5 structure building step) as chain A using Modeller 10.4 [65] to ensure structural completeness. CDKL5 was considered chain A, while the structure of its binding partners was obtained from AlphaFold2 [90], and their chains were designated as Chain B. Upon curation of consensus motif (Table 1) through the literature review, these motif annotations were used to define active interface residues on both interacting proteins. From these, unambiguous distance restraints were generated using the CNS engine in “tbl” format, which specifies direct Cα-Cα contacts between residue pairs. CDKL5 and its binding partners’ dockings were performed to generate a protein–protein complex, where CDKL5 interacts with binding partners around the defined residues in the unambiguous distance restraints file. Afterwards docked complex was used for downstream applications such as binding free energy calculation upon mutation using different available programs.

### 4.5. Binding Free Energy Calculations

In order to assess the impact of *CDKL5* missense variants on protein–protein binding affinity (ΔΔG_binding_) with its binding partners, the following computational methods were employed: SAAMBE-3D, FoldX, mCSM-PPI2, DDMut-PPI, and iSEE.

SAAMBE-3D [91] is a fast, in-house-developed structure-based machine-learning tool that can quantify the change in binding free energy (ΔΔG_binding_) of protein–protein complexes [91]. SAAMBE-3D utilizes 33 knowledge-based features and an XGBoost regression model to predict the ΔΔG_binding_ upon providing the protein–protein complex and a list of mutations. It was trained on the SKEMPI v2.0 dataset while having high predictive accuracy (PCC~0.8).

The iSEE [92] is the interface Structure, Evolution, and Energy-based method, which predicts the impact of mutations on the protein–protein binding free energy (ΔΔG_binding_) by applying a machine-learning framework. It uses a random forest regression model that was trained using the SKEMPI v1.1 dataset, which provides experimentally determined ΔΔG_binding_ values for a wide variety of protein complexes [92]. For each mutation, iSEE builds a feature vector that integrates structural, energetic, and evolutionary information. The structural features include interface energy terms calculated from wild-type complexes using HADDOCK: van der Waals energy (Evdw_wt), electrostatic energy (Eelec_wt), desolvation energy (Edesolv_wt), and buried surface area (BSA_wt). To capture energetic shifts introduced by the mutations, the mutation-induced changes are also included, and these terms are as follows: Evdw_diff, Eelec_diff, Edesolv_diff, and BSA_diff. The mutation-induced energy shifts were calculated as differences between the mutant and wild-type values, using the following formula:X_diff = X_mutant − X_wild-type, where X ∈ {Evdw, Eelec, Edesolv, BSA}

Moreover, the iSEE model utilizes the information from the amino acid sequence, including the original and mutated amino acids (represented with one-hot encoding) and scores that show how conserved each position is across similar proteins, which come from position-specific scoring matrices (PSSMs) [93], including PSSM_wt, PSSM_diff, and PSSMic. To assess the consequence of the *CDKL5* missense variants on protein–protein interactions, we applied a structured pipeline by combining homology modeling, docking, and energy-feature extraction for the preparation of iSEE input data.

In this study, CDKL5 wild-type and mutant structures were first generated using Modeller 10.4 [65], where collected clinical mutations were introduced individually in the experimentally available structure 4BGQ. Upon CDKL5 mutant model generation, the partner protein structures were obtained from AlphaFold2 [90]. Ambiguous interaction restraints were defined using motif-based residue ranges, and both wild-type and mutant complexes were docked with their respective binding partners using the HADDOCK3 program [89]. After docking, HADDOCK3 output energies (Evdw, Eelec, Edesolv, and BSA) were obtained from the top-scoring clusters for both wild-type and mutant complexes. Parallelly, to compute PSSMs for CDKL5, evolutionary conservation profiles were generated using PSI-BLAST [94]. After compiling all energetic and sequence-derived features into a unified feature matrix compatible with iSEE, the trained random forest regression model was applied to predict the ΔΔG_binding_ values for each CDKL5 variant across its respective protein–protein complexes.

### 4.6. CDKL5 Variant Reclassification

*CDKL5* variant reclassification was performed via structure-based ΔΔG_folding_ and ΔΔG_binding_ metrics in kcal/mol, integrating folding stability (ΔΔG_Fmax_) and CDKL5-target protein partner’s binding affinity (ΔΔG_Bmax_) across the CDKL5 kinase domain (residues 1–302). ΔΔG_Fmax_ captured the maximal absolute ΔΔG_folding_ per variant across methods, while ΔΔG_Bmax_ reflected the peak mean ΔΔG_binding_ across four substrate complexes. Empirical thresholds, derived from ClinVar-annotated “benign” and “pathogenic” variants, enable the reassignment of variants. Subsequently, the performance of PolyPhen-2 [95], MutPred2 [96], ESM-1v [76], and AlphaMissense [97] was employed to predict binary pathogenicity labels (benign/pathogenic) and compare them against ClinVar annotations.

## Figures and Tables

**Figure 1 ijms-26-08399-f001:**
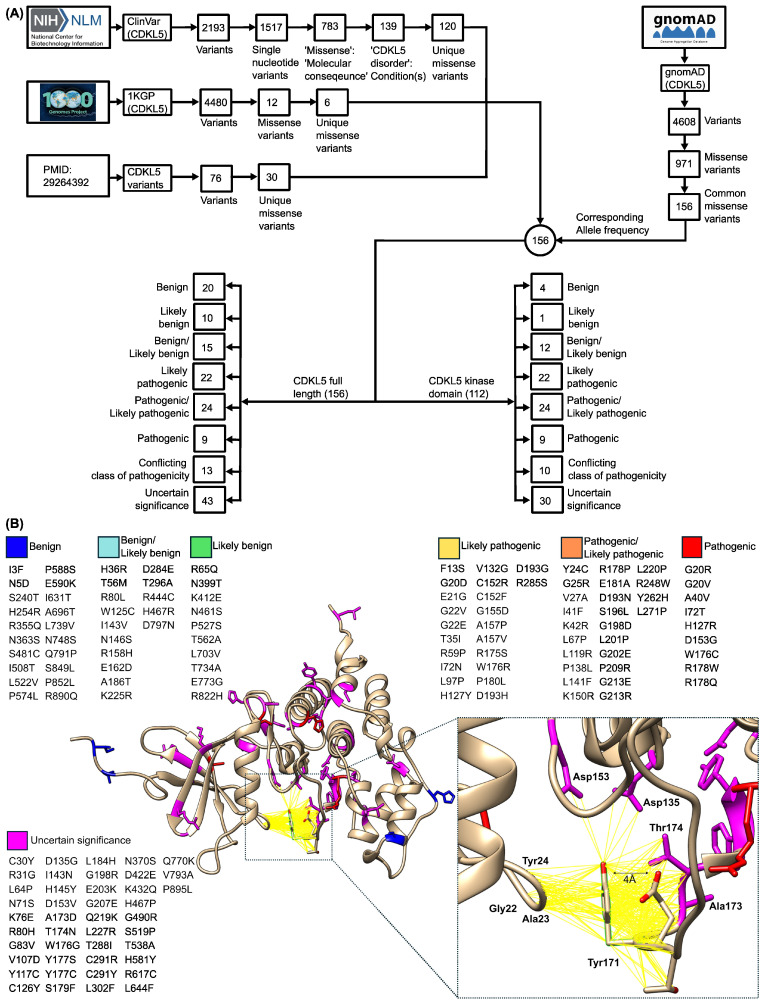
Dataset curation and structural mapping of *CDKL5* missense variants. (**A**) Variant dataset curation. From the ClinVar database of NCBI, the query for “CDKL5” returned 2193 variants. Upon multistep filtering, the unique missense variants were found to be 120. The 1000 Genomes Project (1KGP), another database that hosts variant data from healthy individuals, resulted in 6 unique missense variants for the *CDKL5* gene region. A recent literature review [46] provides 30 unique missense variants. By combining variants from these sources, a unique set of 156 missense variants was assembled, and corresponding allele frequencies were retrieved from the gnomAD database. Among these 156 curated variants, 112 are located within the kinase domain. (**B**) *CDKL5* variants mapped on the kinase domain of the CDKL5 protein. Residues interacting with CDKL5’s Y171 were identified using UCSF Chimera’s “FindClash” tool with a 4 Å threshold (PDB ID: 4BGQ).

**Figure 3 ijms-26-08399-f003:**
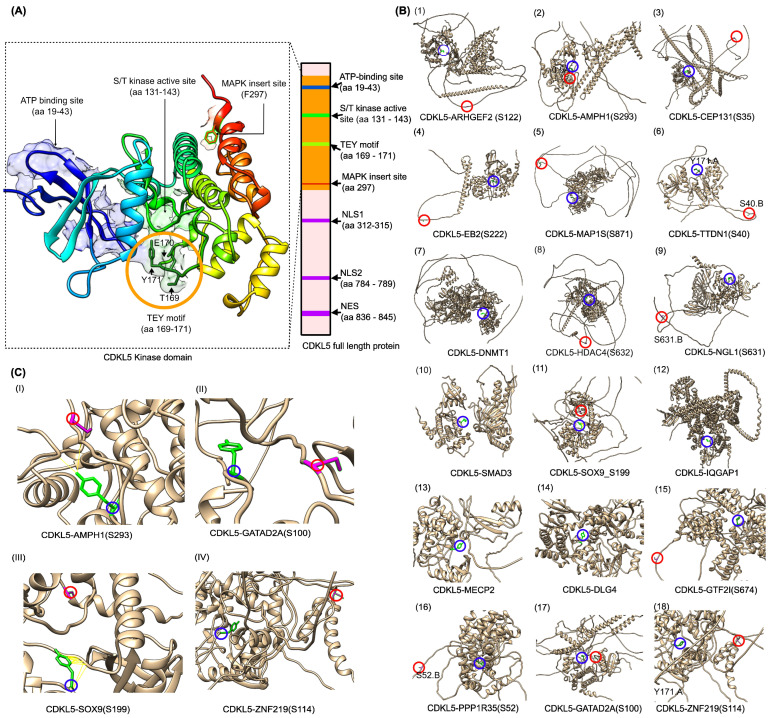
Structural modeling and interaction analysis of CDKL5 kinase domain with predicted substrate complexes. (**A**) Homology model of the CDKL5 kinase domain (residues 1–302) built with Modeller10.4 [65], using PDB ID 4BGQ as the template. (**B**) Predicted CDKL5-target protein complexes for each of the twenty-four collected substrates (some of the complexes were not shown due to low confidence in prediction/error in predictions) using ColabFold (a high-throughput adaptation of AlphaFold-Multimer) [66]. Predicted confidence (pLDDT) scores for CDKL5-partner complexes are provided in the Appendix A. Colored circles indicate key residues: blue highlights the CDKL5 Y171, and red marks highlight the substrate’s phosphorylation site. (**C**) Structural assessment of CDKL5-target protein–protein interactions. (**C**(I)) In the CDKL5-AMPH1(S293) complex, CDKL5 Y171 and substrate phosphosite residue S293 are in close proximity. (**C**(II)) In the CDKL5-GATAD2A(S100) complex, CDKL5 Y171 and GATAD2A S100 seem close enough, but there are no direct interactions. (**C**(III)) In CDKL5-SOX9(S199), CDKL5 Y171 and SOX9’s phosphorylation site S199 also seem close enough. (**C**(IV)) In CDKL5-ZNF219(S114), CDKL5 Y171 and the ZNF219’s phosphopho-site S114 are far from each other.

**Figure 4 ijms-26-08399-f004:**
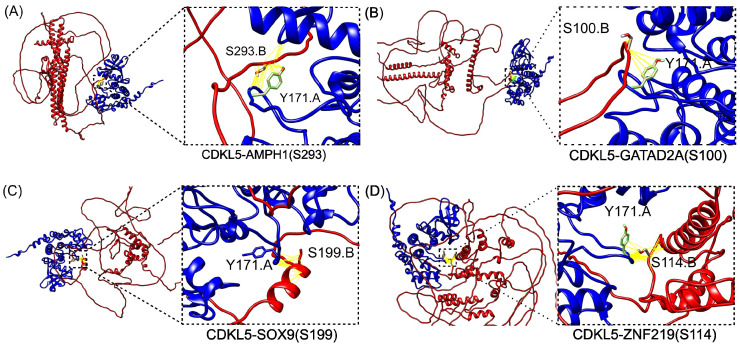
Structural and energetic profiles (Appendix A) of CDKL5-target protein complexes. (**A**–**D**) HADDOCK3-docked models of CDKL5 (blue) bound to four substrates (red) known as phosphorylation targets: (**A**) AMPH1 at S293, (**B**) GATAD2A at S100, (**C**) SOX9 at S199, and (**D**) ZNF219 at S114, where each inset highlights each phosphoserine-positioned hydrogen-bonding proximity (2.7–3.3 Å, yellow lines) to the catalytic residue Y171.A, indicating direct engagement of the active site.

**Figure 5 ijms-26-08399-f005:**
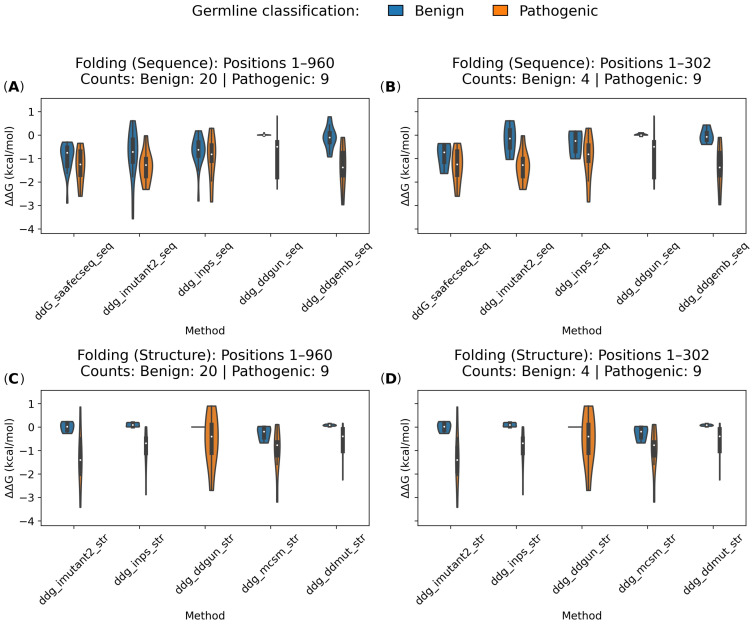
Sequence- and structure-based folding free energy change (ΔΔG_folding_ in kcal/mol) predictions for *CDKL5* missense variants. Violin plots illustrate the distribution of predicted ΔΔG_folding_ (kcal/mol) predictions for pathogenic and benign *CDKL5* variants across five sequence-based (**A**,**B**) and five structure-based (**C**,**D**) computational methods. ΔΔG_folding_ values are plotted for variants located within the full-length protein (residues 1–960) (**A**,**C**) and the kinase domain (residues 1–302; (**B**,**D**). Blue and orange violins represent benign and pathogenic variants, respectively, as classified by germline classification. Sequence-based methods include SAAFEC-SEQ, I-Mutant2.0, INPS, DDGun, DDGemb, and structure-based methods include I-Mutant2.0, INPS, DDGun, mCSM, and DDMut. The figure highlights overall trends in destabilization, with pathogenic variants generally exhibiting more negative ΔΔG_folding_ values, particularly in the kinase domain (1–302) and in predictions from I-Mutant2.0, DDGemb, and mCSM. Among sequence-based methods, I-Mutant2.0, DDGemb, and SAAFEC-SEQ moderately distinguish between benign and pathogenic variants, with the clearest separation observed in the kinase domain (**B**). Structure-based methods such as I-Mutant2.0, mCSM, INPS, and DDMut show even stronger separation, particularly within the kinase domain (**D**). These results indicate that structure-based tools offer superior sensitivity in detecting the destabilizing effects of variants, with I-Mutant2.0 (structure) and mCSM demonstrating the strongest discriminatory performance between pathogenic and benign variants.

**Figure 6 ijms-26-08399-f006:**
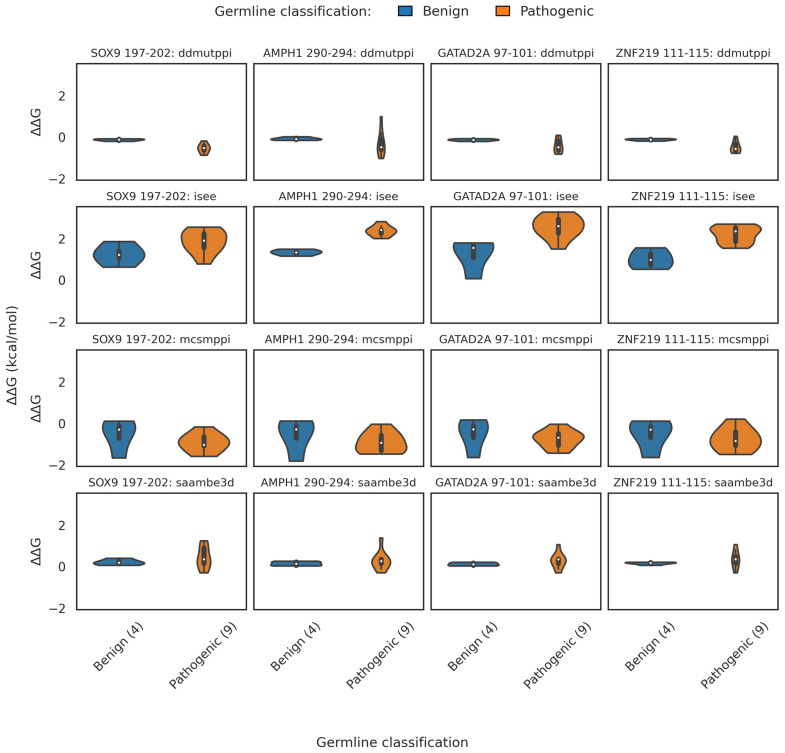
Structure-based ΔΔG_binding_ (in kcal/mol) profiles of CDKL5 kinase-domain variants across binding partners’ phosphosite motifs. Violin plots illustrate predicted changes in binding free energy (ΔΔG_binding_, kcal/mol) for benign (blue) and pathogenic (orange) single-residue variants within the CDKL5 kinase domain (residues 1–302), evaluated at four phosphosite motifs corresponding to known binding partners: SOX9 (phosphomotif: 197–202), AMPH1 (phoshomotif: 290–294), GATAD2A (phosho-motif: 97–101), and ZNF219 (phoshomotif: 111–115). Each row represents one of four structure-based predictors, namely DDMutPPI, iSEE, mCSM-PPI, and SAAMBE-3D, while columns represent the respective CDKL5-target protein complex where the binding partner’s phosphosite regions docked with the CDKL5’s TEY (169–171) motif using the HADDOCK3 program. White circles inside violins indicate median ΔΔG_binding_ values; inner bars denote interquartile ranges. Variant counts for each germline class are shown in brackets at the ends of the corresponding *x*-axis categories. Full summary statistics (n, mean, median, and SD) are available in Appendix A.

**Figure 7 ijms-26-08399-f007:**
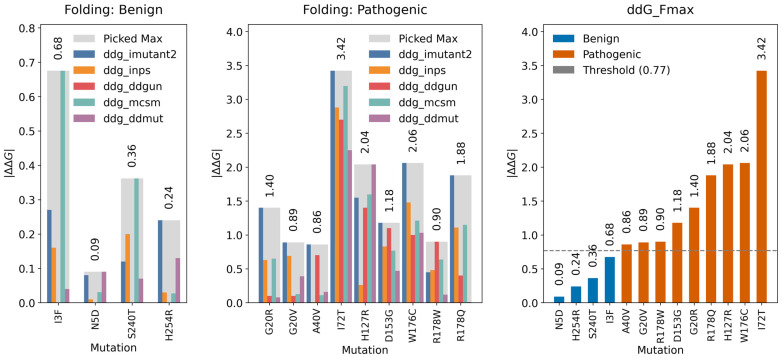
Determination of ΔΔG_folding_ cutoff in kcal/mol to differentiate pathogenic and benign *CDKL5* variants. (**Left**): The bar plots represent the absolute ΔΔG_folding_ for four benign *CDKL5* missense variants. Each colored bar corresponds to a distinct prediction method (I-Mutant2.0, INPS, DDGun, mCSM, and DDMUT), while the gray bar denotes the maximum ΔΔG_folding_ value (ΔΔG_Fmax_) for each variant, with numerical values labeled. Note that DDGun predicted zero kacal/mol ΔΔG_folding_ for all benign variants. (**Middle**): Equivalent visualization for nine pathogenic variants. (**Right**): All thirteen variants are ranked by ΔΔG_Fmax_ and color-coded by germline classification, with blue indicating benign and orange indicating pathogenic. A horizontal dashed line at 0.77 kcal/mol marks the midpoint between the highest benign ΔΔG_Fmax_ value (0.68 kcal/mol) and the lowest pathogenic ΔΔG_Fmax_ value (0.86 kcal/mol), defining an optimal threshold for variant discrimination based on ΔΔG_folding_.

**Figure 8 ijms-26-08399-f008:**
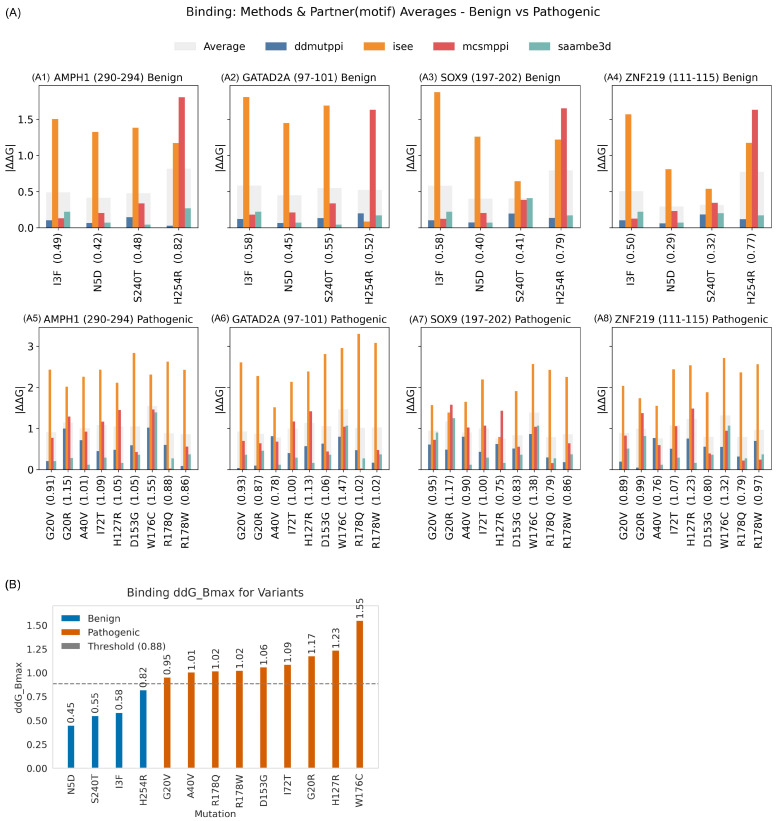
For each variant, the binding free energy changes (ΔΔG_binding_) in kcal/mol were computed using averaged values across CDKL5 binding partners, and the ΔΔG_Bmax_ threshold was subsequently determined. (**A**) Multi-panel bar charts of absolute ΔΔG_binding_ for CDKL5-target protein complexes at their consensus phosphosite motifs. (**A1**–**A4**) Binding free energy changes (ΔΔG_binding_) due to benign variants. (**A5**–**A8**) Changes in ΔΔG_binding_ due to pathogenic variants for (**A1**,**A5**) CDKL5-AMPH1 (motif 290–294), (**A2**,**A6**) CDKL5-GATAD2A (motif 97–101), (**A3**,**A7**) CDKL5-SOX9 (motif 197–202), and (**A4**,**A8**) CDKL5-ZNF219 (motif 111–115). In each panel, opaque colored bars show the mean |ΔΔG_binding_| across methods for each variant. Variant labels include this complex average across methods in parentheses (e.g., “I3F (0.58)”). Legends above the panels represent the type of bars (methods and average). (**B**) Sorted bar plot of ΔΔG_Bmax_: the maximum complex average |ΔΔG_binding_| across the four CDKL5-target protein motifs for all thirteen variants (four benign and nine pathogenic). Blue bars denote benign; orange bars denote pathogenic. Numerical ΔΔG_Bmax_ values are labeled above each bar. The dashed gray line indicates the classification cutoff (0.88 kcal/mol), defined as the midpoint between the highest benign ΔΔG_Bmax_ (0.82 kcal/mol) and the lowest pathogenic ΔΔG_Bmax_ (0.95 kcal/mol).

**Figure 9 ijms-26-08399-f009:**
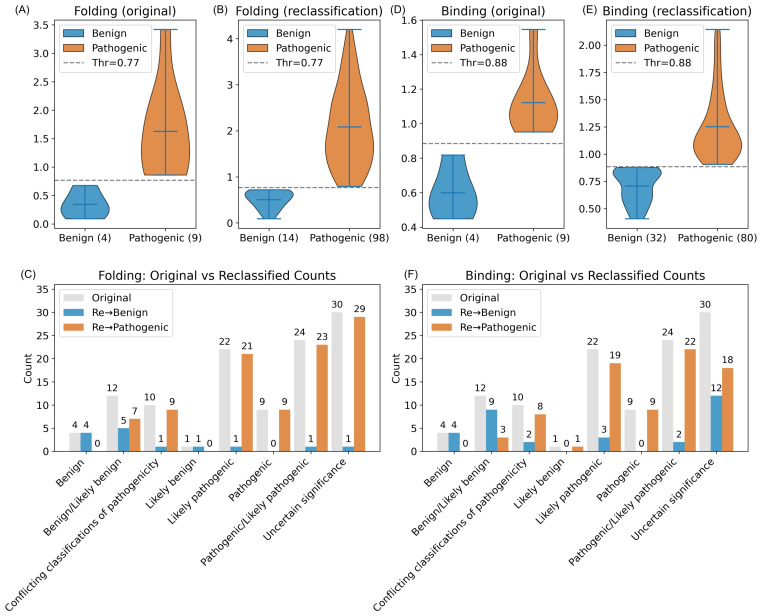
Changes in folding (ΔΔG_folding_) and binding (ΔΔG_binding_) in kcal/mol free energies provided mechanistic insights that supported the reclassification of variant pathogenicity. (**A**,**B**,**D**,**E**) Violin plots of ΔΔG_folding_ (**A**,**B**) and ΔΔG_binding_ (**D**,**E**) distributions, both before (**A**,**D**) and after (**B**,**E**) reclassification of variants. The *x*-axes label the group (benign and pathogenic), with sample size in parentheses. Blue violins (**left**) are benign, orange violins (**right**) are pathogenic, and the dashed gray line marks the classification thresholds. (**C**,**F**) Grouped-bar charts summarizing original versus reclassified counts for folding (**C**) and binding (**F**). In them, the light gray bars show the original germline classification counts, blue bars are variants reclassified as benign, and orange bars are variants reclassified as pathogenic.

**Figure 10 ijms-26-08399-f010:**
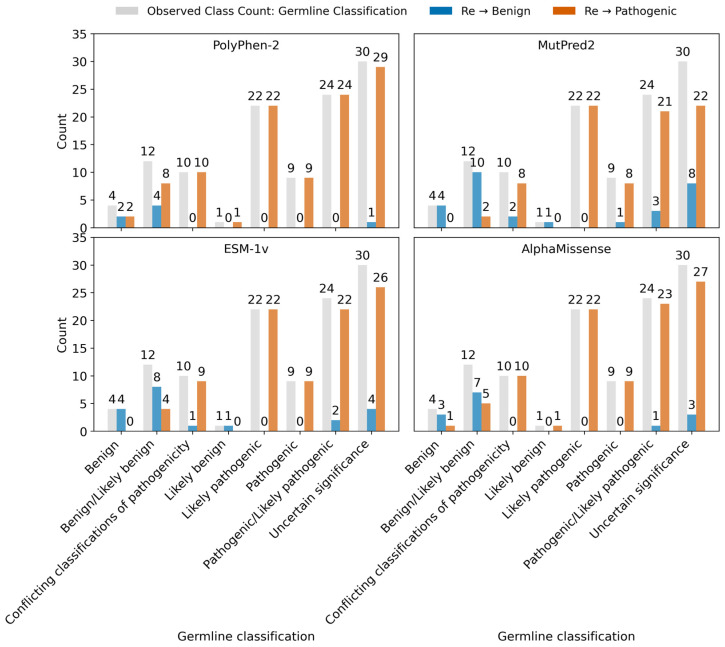
Comparative reclassification of *CDKL5* germline variants by four pathogenicity predictors. Each panel represents the initial distribution of *CDKL5* germline variant classification (gray bars), including benign, benign/likely benign, conflicting classifications of pathogenicity, likely benign, likely pathogenic, pathogenic/likely pathogenic, and uncertain significance. These are compared with how each computational tool reassesses the same variants as either benign (blue) or pathogenic (orange). The four panels, shown clockwise from the top left, represent outputs from PolyPhyn-2, Mutpred2, ESM-1v, and AlphaMissense. Numeric labels above each bar indicate variant counts per category.

**Figure 11 ijms-26-08399-f011:**
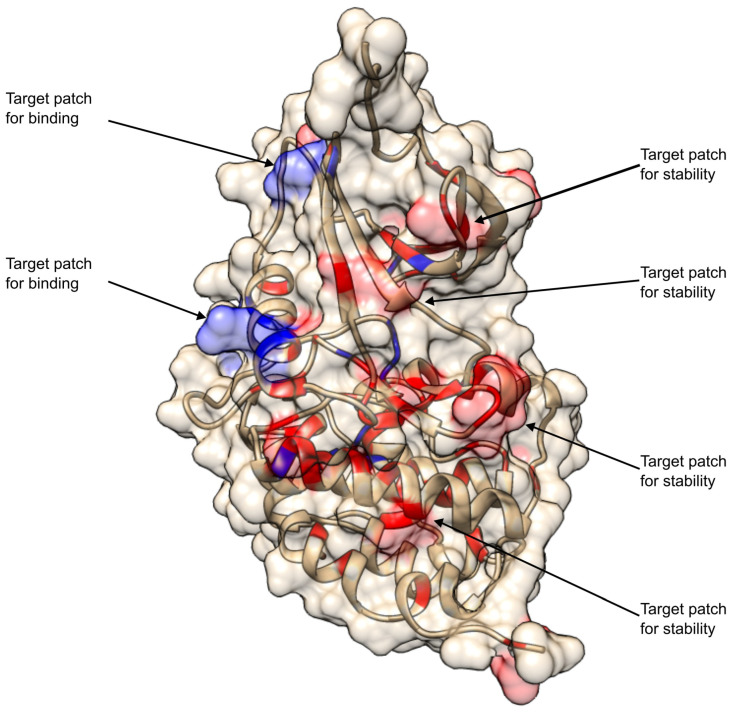
Structural mapping of thermodynamically reclassified pathogenic *CDKL5* variants within the kinase domain. Thermodynamically reclassified pathogenic variants are mapped on the CDKL5 kinase domain based on their relative impact on protein folding and binding stability. Variants exhibiting higher folding destabilization than binding destabilization (ΔΔG_Fmax_ > ΔΔG_Bmax_) are annotated in red with transparent-surface rendering. Conversely, variants where binding destabilization is equal to or exceeds folding destabilization (ΔΔG_Fmax_ ≤ ΔΔG_Fmax_) are annotated in blue.

## Data Availability

Code used in the analysis for this project is deposited in this repository: https://github.com/paulshamrat/cdkl5-variants (accessed on 15 August 2025).

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
