# Peer review of "CDKL5 Deficiency Disorder: Revealing the Molecular Mechanism of Pathogenic Variants"

_ijms, 2025, doi:10.3390/ijms26178399_

Round 1

Reviewer 1 Report

Comments and Suggestions for Authors

The manuscript provides comprehensive investigation of the molecular effects of variants in a particular gene, the CDKL5 gene, which are found in the human population. The work enriches the list of known mutations and adds to the list of partners four CDKL5 interacting targets. It is demonstrated that there is a distinctive difference between the effect of pathogenic vs benign variants on the folding and binding free energy of CDKL5 protein. It is shown that pathogenic mutations greatly affect these thermodynamics quantities, which is used to re-classify variants with unknown significance. The paper not only reveals the molecular mechanism of pathogenic mutations in CDKL5 protein, but suggest plausible directions for developing therapeutic solutions. The senior the author, Alexov, is a top researcher in the field. I recommend the publication after minor changes. 

Major comments:

None

Minor comments:

1)      OMIM database must be cited.

2)      What is the meaning of “*” in the following expression “By using a patient-derived male iPSC* cell line [CDKL5; c.175C > T, p.(Arg59*)] al….”

3)      The first letter of the words in subtitles is sometimes capitalized, other times it is not. Should be made consistent.

4)      What is the motivation or the rationale for selecting ABS(DDG) for the threshold instead of taking into account the sign of the DDG change? This must be clarified.

5)      I would like to suggest that the paragraph “2.3.6. Variants reclassification based on pathogenicity score” be expanded to clearly indicate the overall failure of the evolution-based methods to correctly predict pathogenic and benign variants and thus to justify the approach presented in the paper. 

Author Response

We thank the reviewer for the useful comments and suggestions. They were carefully considered and appropriate corrections were made. Below we outline point-by-point the responses to the reviewer's sugegstions:

1)      OMIM database must be cited.

The OMIM database has been cited in the first line of the Introduction just after the OMIM entry. The OMIM reference is as follows: Amberger, J.S.; Bocchini, C.A.; Scott, A.F.; Hamosh, A. OMIM.org: leveraging knowledge across phenotype-gene relationships. Nucleic Acids Res 2019, 47, D1038–D1043, doi:10.1093/nar/gky1151.

2)      What is the meaning of “*” in the following expression “By using a patient-derived male iPSC* cell line [CDKL5; c.175C > T, p.(Arg59*)] al….”

We thank the reviewer for raising this point. We revised the sentence in the main text to read: “By using a patient-derived male iPSC cell line carrying the CDKL5 variant c.175C>T (resulting in p.Arg59*, where ‘*’ denotes a premature stop codon-according to HGVS nomenclature) [44]….” Here, the asterisk “*” follows HGVS guidelines and indicates a nonsense mutation leading to a premature termination codon.

3)      The first letter of the words in subtitles is sometimes capitalized, other times it is not. Should be made consistent.

We thank the reviewer for noticing this. The formatting of all subtitles has now been revised so that capitalization is consistent throughout the manuscript.

4)      What is the motivation or the rationale for selecting ABS(DDG) for the threshold instead of taking into account the sign of the DDG change? This must be clarified.

In previous works we have demonstrated that any large deviation of DDG, making the protein more or less stable (Hum Mol Genet 2012 Oct 15;21(20):4497-507.; Curr Opin Struct Biol 2023 Jun:80:102572), is associated with pathogenicity. This was the rational for applying ABS(DDG) instead of taking into account the sign of the DDG change.

5)      I would like to suggest that the paragraph “2.3.6. Variants reclassification based on pathogenicity score” be expanded to clearly indicate the overall failure of the evolution-based methods to correctly predict pathogenic and benign variants and thus to justify the approach presented in the paper. 

We thank the reviewer for pointing out the need for clearer justification of section 2.3.6. We have revised the section to define the accuracy metric (Benign vs. non-Benign grouping) included in the Supplementary Table 6 summarizing performance across predictors and expanded the text in the manuscript to report both overall accuracy and strict correctness on the 4 benign and 9 pathogenic variants. This clarification highlights that although one of them, the ESM-1v, correctly identified all strictly labeled benign and pathogenic cases, all pathogenicity predictor methods reassign uncertain/conflicting variants as pathogenic. These inconsistencies support our rationale for adopting thermodynamics-based reclassification.

Reviewer 2 Report

Comments and Suggestions for Authors

Summary
This manuscript provides a systematic analysis of 156 CDKL5 missense variants, integrates structural modeling of CDKL5–substrate complexes, and applies ΔΔG predictions to propose thresholds that separate pathogenic from benign variants. The study is timely, mechanistically interesting, and of potential clinical utility.

Strengths

  • Clear curation and mapping of variants.

  • Focused mechanistic framing on the TEY motif (Y171).

  • Transparent reporting of ΔΔG thresholds.

  • Helpful comparison with established variant prediction tools.

Comments for Authors (minor, at your discretion)

  1. Docking distances – The text mentions hydrogen bonds at ≤1 Å, which is shorter than typical donor–acceptor distances. You may wish to clarify or adjust the description.

  2. Performance validation – The paper would be strengthened by reporting formal metrics (AUROC, sensitivity/specificity) or cross-validation, though the illustrative comparisons are already useful.

  3. ΔΔG tool outputs – Some tools are described as classification-only yet reported quantitatively; clarifying this would aid reproducibility.

  4. Threshold robustness – The cutoff derivation could be supported with simple cross-validation or external validation.

  5. Docking/model confidence – Providing model-confidence measures (e.g., pLDDT, HADDOCK scores) would add transparency.

  6. Interaction partners – A table categorizing partners by evidence level (biochemical, proteomic, motif only) would be a nice addition.

Minor suggestions

  • Correct a few typographical errors (e.g., “protien,” “CDLK5”).

  • Provide tool versions, parameters, and seeds for reproducibility.

  • Consider linking reclassification recommendations to ACMG/AMP criteria to illustrate practical clinical use.

Recommendation
The manuscript is well executed and of broad interest. I recommend acceptance pending minor clarifications and improvements at the authors’ discretion.

Author Response

We thank the reviewer for the useful comments and suggestions. They were carefully considered and appropriate changes were made. Below we outline point-by-point the changes made in response to reviewer's suggestions:

Comments for Authors (minor, at your discretion)

  1. Docking distances – The text mentions hydrogen bonds at ≤1 Å, which is shorter than typical donor–acceptor distances. You may wish to clarify or adjust the description.

We thank the reviewer for catching this mistake. We had inadvertently written “≤1 Å” in the text. Based on our docking analysis, the closest Tyr171 substrate motif distances fall in the 2.7-3.3 Å range. We have corrected it in the manuscript.

  1. Performance validation – The paper would be strengthened by reporting formal metrics (AUROC, sensitivity/specificity) or cross-validation, though the illustrative comparisons are already useful.

Thanks for the suggestion. We have added Supplementary Figure 2, where we used label-flip-invariant metrics (AUROC_sym, Balanced_Accuracy_sym, MCC_sym) instead of cross-validation to account for the limited dataset size with class imbalance.

  1. ΔΔG tool outputs – Some tools are described as classification-only yet reported quantitatively; clarifying this would aid reproducibility.

All tools used for predicting ΔΔGbinding or ΔΔGfolding provided ΔΔG values, which we incorporated. If any tool also offered direct classification, we did not use it, as our classification approach is based on score aggregation, as detailed in the Methods section.

  1. Threshold robustness – The cutoff derivation could be supported with simple cross-validation or external validation.

We agree with the reviewer that cross-validation would have been ideal. However, the limited dataset size (4 benign and 9 pathogenic) does not allow for any n-fold cross-validation because the validation set in such a case would be too small (~2-3 data points), and then we have to account for class imbalance as well. So, we would end up with no meaningful cross-validation set. Considering these points, we decided against the cross-validation.

  1. Docking/model confidence – Providing model-confidence measures (e.g., pLDDT, HADDOCK scores) would add transparency.

We thank the reviewer for pointing this out. The HADDOCK3 Score is already provided in Supplementary Table 3: HADDOCK3 Per‐Model Energy Component Breakdown for CDKL5–Substrate Complexes. Also, pLDDT score has been added as Supplementary Table 7: Predicted confidence (pLDDT) scores for CDKL5-partner complexes, as well as pLDDT plots are presented for the CDKL5-partner complex on Supplementary Figure 3.

  1. Interaction partners – A table categorizing partners by evidence level (biochemical, proteomic, motif only) would be a nice addition.

We thank the reviewer for suggesting this. We have added a dedicated column named “Evidence” on Table 1, which explicitly mentions if the binding partner is identified by motif, biochemically, or both.

Minor suggestions

  • Correct a few typographical errors (e.g., “protien,” “CDLK5”): Thank you for spotting these errors! We have fixed these typos
  • Provide tool versions, parameters, and seeds for reproducibility. We have provided these versions, parameters, and seeds where required.
  • Consider linking reclassification recommendations to ACMG/AMP criteria to illustrate practical clinical use.

We have now mapped the concordant in silico reclassification results to ACMG/AMP computational criteria at supporting strength. Specifically, variants for which both ΔΔGfolding and ΔΔGbinding indicate pathogenic effect are annotated as PP3 (supporting), whereas variants for which both indicate benign effect are annotated as BP4 (supporting). Discordant cases are not assigned computational criteria. We emphasize that these codes represent supporting-level evidence only and are not substitutes for PS3/BS3 functional criteria. The result is presented in Supplementary Table 8.